# Evaluation of the Association between *FGFR2* Gene Polymorphisms and Breast Cancer Risk in the Bangladeshi Population

**DOI:** 10.3390/genes14040819

**Published:** 2023-03-29

**Authors:** Nusrat Jahan, Mobashera Begum, Md Abdul Barek, Md. Abdul Aziz, Md. Shafiul Hossen, Khokon Kanti Bhowmik, Tahmina Akter, Md. Rabiul Islam, Hadi Sajid Abdulabbas, Mohammad Safiqul Islam

**Affiliations:** 1Department of Pharmacy, Faculty of Science, Noakhali Science and Technology University, Noakhali 3814, Bangladesh; 2Laboratory of Pharmacogenomics and Molecular Biology, Department of Pharmacy, Noakhali Science and Technology University, Noakhali 3814, Bangladesh; 3Department of Pharmacy, University of Asia Pacific, Dhaka 1205, Bangladesh; 4Continuous Education Department, Faculty of Dentistry, University of Al-Ameed, Karbala 56001, Iraq; 5Bangladesh Pharmacogenomics Research Network (BdPGRN), Noakhali 3814, Bangladesh

**Keywords:** breast cancer, *FGFR2*, polymorphisms, genotype, haplotype, PCR-RFLP

## Abstract

Breast cancer is considered the most frequent cause of mortality from malignancy among females. Fibroblast growth factor receptor 2 (*FGFR2*) gene polymorphisms are highly related to the risk of breast cancer. However, no investigation has been carried out to determine the association of *FGFR2* gene polymorphisms in the Bangladeshi population. Based on polymerase chain reaction–restriction fragment length polymorphism (PCR-RFLP), this study was performed to evaluate the association of *FGFR2* (rs1219648, rs2420946, and rs2981582) variants in 446 Bangladeshi women (226 cases and 220 controls). A significant association of the *FGFR2* rs1219648 variant with breast malignancy was reported in additive model 1 (aOR = 2.87, *p* < 0.0001), additive model 2 (aOR = 5.62, *p* < 0.0001), the dominant model (aOR = 2.87, *p* < 0.0001), the recessive model (aOR = 4.04, *p* < 0.0001), and the allelic model (OR = 2.16, *p* < 0.0001). This investigation also explored the significant association of the rs2981582 variant with the risk of breast cancer in additive model 2 (aOR = 2. 60, *p* = 0.010), the recessive model (aOR = 2.47, *p* = 0.006), and the allelic model (OR = 1.39, *p* = 0.016). However, the *FGFR2* rs2420946 polymorphism showed no association with breast cancer except in the overdominant model (aOR = 0.62, *p* = 0.048). Furthermore, GTT (*p* < 0.0001) haplotypes showed a correlation with breast cancer risk, and all variants showed strong linkage disequilibrium. Moreover, in silico gene expression analysis showed that the *FGFR2* level was upregulated in BC tissues compared to healthy tissues. This study confirms the association of *FGFR2* polymorphisms with breast cancer risk.

## 1. Introduction

Breast cancer is one of the most prevalent types of malignancy. The number of newly diagnosed patients identified was more than two million, approximately 11.7% of total cancer cases worldwide, in 2020. According to a report by GLOBOCAN, it is the fifth leading cause of morbidity (684,996 deaths, 6.9%) in the world [1]. It is also considered the most common cancer in 159 nations out of 185 and the most common cause of death in 110 countries. In addition, breast cancer is gradually becoming a major concern in South Asian countries, including Bangladesh, where the incidence rate is 22.5 per 100,000 women [2]. According to prior research, limited awareness of and learning about breast cancer among the people of Bangladesh has increased the risk of developing breast cancer [3]. Several factors, including maturity, circulating hormone concentrations, hereditary factors, and environmental exposure, contribute to the susceptibility to BC; however, the exact reason behind this is unclear. Moreover, hereditary factors are considered the most crucial attributes in the genetic variation in susceptible genes of approximately 5–10% of all breast cancer patients [4]. In addition, genome-wide association studies have found that numerous breast cancer-related genes and single-nucleotide polymorphisms (SNPs) are linked to the occurrence and progression of breast cancer [5].

Fibroblast growth factor receptor 2 (*FGFR2*) is a membrane-associated receptor tyrosine kinase responsible for transmitting a signal from fibroblast growth factors (FGFs). It is mapped on chromosome 10q26 and has a minimum of 22 members that serve numerous functions in cell proliferation and differentiation, growth, survival, and cell death. However, mutation of this gene leads to the proliferation and survival of tumor cells but also could suppress tumor growth [6,7,8,9,10]. In addition, *FGFR2* mutations enhance DNA damage signaling and p53-induced senescence [11]. Oncogenic *FGFR2* triggering is induced by *FGFR2* missense mutations surrounding the third Ig-like domain. In contrast, those interiors in the tyrosine kinase domain generate oncogenic *FGFR2* activation owing to the acquisition of ligand sovereignty [12].

FGFR2 may be a plausible mediator of breast cancer risk due to its role in signaling, the incidence of *FGFR2* gene amplification in breast carcinoma, and the presence of risk SNPs within its intron 13. Intron 2 of the *FGFR2* gene, highly conserved in mammals, has numerous possible transcription factor binding sites, some of which are positioned adjacent to the variants and speculated to be most strongly related to breast cancer [13,14,15]. There is also evidence that the risk allele contributes to an upregulation in the expression of *FGFR2* [16]. Therefore, the *FGFR2* gene has been identified as a potential risk factor for breast cancer development due to the genetic variations in this gene. The *FGFR2* gene polymorphisms (rs1219648 A > G, rs2420946 C > T, and rs2981582 C > T) are located on intron 2 of chromosome 10 at different positions. SNP rs1219648 A > G is located in Chr10:121586676, whereas rs2420946 C > T is positioned in Chr10:121591810 and rs2981582 C > T is mapped in Chr10:121592803. These variants are predicted to enhance histone marks and motif changing. In addition to these functions, rs1219648 A > G and rs2981582 C > T are also speculated to be involved in GRASP QTL hitting. Previous studies also frequently showed that rs1219648, rs2420946, and rs2981582 are linked with the development of breast cancer in different ethnicities [12,14,15,17,18]. However, their association with breast cancer is still ambiguous due to the differences in ethnicity, geographic characteristics, and other determinants.

The associations of these polymorphisms were found to be inconsistent in different ethnic groups; however, there is no such investigation on the Bangladeshi population. Considering this fact, the current case–control study was performed to clarify the potential relationship between breast cancer and three common *FGFR2* polymorphisms (rs1219648, rs2420946, and rs2981582).

## 2. Materials and Methods

### 2.1. Study Population

This case–control study was conducted at the Pharmacogenomics and Molecular Biology Laboratory at Noakhali Science and Technology University, Bangladesh, and approved by the ethical committee of the National Institute of Cancer Research and Hospital (NICRH), Bangladesh (NICRH/Ethics/2019/446). Around 446 women participated in this study, including 226 breast cancer patients and 220 healthy individuals. A comprehensive questionnaire was used to collect patients’ demographic and clinicopathological data with informed written consent. Selection guidelines for healthy controls mentioned no indication of cancer and other diseases such as kidney, liver, and lung diseases. For the clinicopathological features of breast cancer patients, we mostly focused on age, marital status, BMI, histological type of cancer, progesterone receptor status, estrogen receptor status, human epidermal growth factor receptor 2 (HER2/neu) status, grade, tumor measurement, lymph node condition, nodal stage, and distant metastasis. All information was retrieved from the patient’s medical record with the assistance of a physician. The Helsinki Declaration and its further correction were followed during the study [19].

### 2.2. Sample Preparation

Approximately 3 mL of blood was collected from all participants and stored in an EDTA-Na_2_-containing sterile tube at −80 °C until DNA extraction. DNA extraction was conducted via an established method routinely used in our lab [20]. Genotyping of all extracted DNA was performed by PCR using the target DNA fragment. A micro-volume spectrophotometer measured the purity and concentration of all DNA at 260 nm and 280 nm (Genova Nano, Jenway, IL, USA). Primer Blast online-based software was utilized to design the primer sequences.

### 2.3. Genotyping

SNPs rs1219648, rs2420946, and rs2981582 were genotyped by using the PCR-based restriction fragment length polymorphism (RFLP) technique. A working mix was prepared for each sample by adding EmeraldAmp GT PCR Master Mix (2x) and two designed complementary primers (forward and reverse) at a suitable concentration. For PCR, 20 µL of this working mix and 1 µL of DNA sample were mixed in a PCR tube and then amplified at certain conditions. The gel electrophoresis (1% agarose) method was followed to analyze the desired PCR fragments. Digestion of PCR products of all three SNPs was performed with HinP1I (Thermo Fisher Scientific, MA, USA) by incubation at 37 °C overnight. Digestion conditions and expected fragment lengths of all SNPs are listed in Table 1. Finally, the digested fragments were stained with Ethidium Bromide (EtBr) and visualized on 1.3% agarose gel electrophoresis, where a 100 bp DNA ladder was used to measure the fragments more accurately. The reverse and forward primer sequences and their required PCR conditions with fragment lengths are given in Table 1 (Appendix A)

### 2.4. Statistical Analysis

The chi-square (χ^2^) test was used to evaluate the Hardy–Weinberg Equilibrium (Appendix A), and an odds ratio (OR) with 95% confidence interval (CI) was used to examine the risk of breast cancer. A significant association was considered at *p* < 0.05. ORs were adjusted (aOR) for BMI, age, marital status, family history, smoking status, OCP history, age at menarche, age of menopause, and drinking status for all genetic models except the allele model. In addition, targeted variants were compared with several clinicopathological characteristics. Meanwhile, the *p*-value was corrected by using Bonferroni correction, where the significance level was measured at *p* < 0.017 [21,22]. Additionally, linkage disequilibrium (LD) and haplotypes of rs1219648, rs2981582, and rs2420946 for breast cancer risk were also determined by applying the SHEsis online application.

## 3. Results

### 3.1. Demographic and Clinicopathological Characteristics

Table 2 shows the detailed demographic and clinicopathological variables. As per the observation, breast cancer was more prevalent among married patients (97.35%) and those aged more than 35 years (69.47%). A slightly significant difference (*p* = 0.049) was observed between cases and controls in terms of family history. Most subjects were overweight (28.76) or obese (30.51%). In addition, invasive ductal carcinoma (57.08%) and grade 2 (57.96%) types of cancer were the most common among the participants. Moreover, a higher percentage of them were recognized with a positive nodal status (64.16%) and Mx metastasis condition (75.22%).

### 3.2. Association of FGFR2 Polymorphisms with Breast Cancer

Table 3 provides the genotype and haplotype-based association of the *FGFR2* rs1219648, rs2420946, and rs2981582 polymorphisms with the risk of breast cancer in the Bangladeshi population. As presented, this study identified a greater percentage of homozygote genotypes of the *FGFR2* rs1219648 variant in controls. In contrast, a higher percentage of heterozygotes and mutant homozygotes of this SNP was observed in patients. As a result, a significant association of these genotypes with breast carcinoma was identified in various association models, including additive model 1 (AG vs. AA: aOR = 2.87, 95% CI = 1.76–3.69, *p* < 0.0001), additive model 2 (GG vs. AA: aOR = 5.62, 95% CI = 2.52–12.54, *p* < 0.0001), the dominant model (AG + GG vs. AA: aOR = 2.87, 95% CI =1.76–4.69, *p* < 0.0001), the recessive model (GG vs. AA + AG: aOR = 4.04, 95% CI = 1.90–8.59, *p* = 0.0001), and the allelic model (A vs. G: OR = 2.16, 95% CI = 1.62–2.89, *p* < 0.0001). Unlike the first variant discussed above, a greater percentage of heterozygotes of the *FGFR2* rs2420946 variant was observed in controls. However, no substantial relationship was observed in any genetic inheritance models for this polymorphism except the overdominant model (CT vs. CC + TT: aOR = 0.62, 95% CI= 0.39–1.0, *p* = 0.048). Again, mutant homozygotes of the rs2981582 variant were more prevalent in cases compared to healthy individuals. As is presented, this study illustrated the significant association of this SNP with an elevated risk of breast cancer in additive model 2 (TT vs. CC: aOR = 2.60, 95% CI = 1.25–5.37, *p* = 0.010), the recessive model (TT vs. CC + CT: aOR = 2.47, 95% CI = 1.13–4.69, *p* = 0.006), and the allelic model (C vs. T: OR = 1.39, 95% CI = 1.06–1.82, *p* = 0.016).

Furthermore, the haplotype-based analysis revealed a significant association of the GTT, ATT, and ATC haplotypes with breast cancer. The GTT haplotype (OR = 2.51, 95% CI = 1.86–3.55, *p* < 0.0001) depicted an enhanced risk of breast cancer, whereas the ATT (OR = 0.18, 95% CI = 0.10–0.33, *p* < 0.0001) and ATC (OR = 0.13, 95% CI = 0.04–0.43, *p =* 0.0009) haplotypes showed a protective association. Moreover, this study also illustrated the linkage disequilibrium (LD) of all SNPs in the breast cancer cases and controls (Figure 1) that represented the LD for rs1219648 and rs2420946 (D’ = 0.931, r^2^ = 0.624); rs1219648 and rs2981582 (D’ = 0.777, r^2^ = 0.389), and rs2420946 and rs2981582 (D’ = 0.823, r^2^ = 0.606).

### 3.3. Association of FGFR2 Polymorphisms with Clinicopathological Variables

We investigated the linkage of breast cancer risk with *FGFR2* gene polymorphisms considering various clinicopathological variables of patients, as illustrated in Table 4. A significant association of the *FGFR2* rs1219648 variant with BC risk was found for the HER2 (+) status, grade 2 cancer, and positive lymph node status of patients. The genotype distribution of this variant increased the breast cancer risk in patients with HER2-positive status (OR = 2.13, 95% CI = 1.10–4.13, *p* = 0.025); however, it showed a protective association among those who had grade 2 cancer (OR = 0.38, 95% CI = 0.19–0.75, *p* = 0.006) and a positive lymph node status (OR = 0.55, 95% CI = 0.30–0.98, *p* = 0.043). In addition, patients identified with N1 nodal status showed a link with breast cancer (OR = 2.64, 95% CI = 1.42–4.93, *p* = 0.002) compared to the N0 nodal status of patients in terms of the rs2420946 polymorphism. Moreover, a protective correlation was also observed in patients with a positive lymph node status (OR = 0.40, 95% CI = 0.22–0.75, *p* = 0.004) in the case of the rs2981582 variant.

### 3.4. Analysis of In Silico Gene Expression

We used the GEPIA (http://gepia.cancer-pku.cn/, accessed on 20 November, 2022) and UALCAN (http://ualcan.path.uab.edu/, accessed on 20 November, 2022) databases to examine the *FGFR2* mRNA expression levels in breast cancer. Breast cancer tissues had considerably more significant levels of *FGFR2* mRNA expression than normal tissues (*p* < 0.01) (Figure 2). We also found significantly low expression of *FGFR2* in Asians (Normal vs. Asian: *p* = 2.83 × 10^−4^) compared to Caucasians (Normal vs. Caucasian: *p* = 8.79 × 10^−1^). On the other hand, moderate expression was observed in African Americans (Normal vs. African American: *p* = 2.09 × 10^−12^) (Figure 3). All three races were compared with healthy individuals from the UALCAN database.

### 3.5. Genotype-Based FGFR2 mRNA Expression

The analysis of genotype-based *FGFR2* mRNA expression for the studied SNPs (rs1219648, rs2420946, and rs2981582) from the Genotype-Tissue Expression (GTEx) database (http://www.gtexportal.org/, accessed on 20 November, 2022) is shown in Figure 4. The eQTL plots show that there was a statistically non-significant difference in the expression level of the *FGFR2* rs1219648 polymorphism in breast tissues (*p* = 0.20) depending on the genotype. In the case of the *FGFR2* rs2420946 and *FGFR2* rs2981582 polymorphisms, the expression levels in breast tissues were also reported to be non-significant between the genotypes (*p* = 0.18 and *p* = 0.26, respectively).

## 4. Discussion

This case–control study was performed to clarify the association between breast cancer and three common *FGFR2* polymorphisms (rs1219648, rs2420946, and rs2981582) in the Bangladeshi population. This is the first study in Bangladesh that has looked at the association of *FGFR2* polymorphisms with breast cancer and observed a significant outcome. The increased incidence of cancer in recent years has had a devastating impact on the physical, mental, and social lives of human beings, making it one of the major problems of the century. The most common type of cancer in women worldwide is breast cancer, and this alarming condition has been associated with the most cancer-related deaths among women [23]. This is increasing day by day, and patients are being diagnosed at an earlier age. Though several causes have been identified behind this cancer, lifestyle and genetic variations are addressed as the main causes among different races. Previous investigations showed that genetic variations in many genes play a crucial role in breast cancer advancement [24]. However, the association between genetic risk factors in the etiology of breast carcinoma is still obscure [25,26].

The *FGFR2* gene belongs to the tyrosine kinase receptor family, which is involved in multiple signaling pathways in tumorigenesis through apoptosis, differentiation, and cell growth [27]. Various studies revealed their relevance to breast cancer risk, where a polymorphism occurred in the *FGFR2* intron 2 location and modulated the interaction of two transcription factors named Oct-1/Runx2 and C/EBPb. Consequently, increased *FGFR2* gene expression in both cell lines and breast tissues occurred [16]. As is observed, the present study also found a significant association of the *FGFR2* rs1219648 polymorphism with breast cancer risk in all determined genetic models, including additive model 1 (aOR = 2.87, *p* < 0.0001), additive model 2 (aOR = 5.62, *p* < 0.0001), the dominant model (aOR = 2.87, *p* < 0.0001), the recessive model (OR = 4.04, *p* = 0.0001), and alleles (OR = 2.16, *p* < 0.0001). Studies conducted in different ethnic and regional populations, such as North Indian, American, Turkish, Arab, and Kazakhstani populations, also revealed the association of rs1219648 with BC risk [28,29,30].

In addition, our study demonstrated the significant relationship between the rs2981582 variant and breast cancer in terms of some genetic models, including additive model 2 (aOR = 2.60, *p* = 0.010), the recessive model (aOR = 2.47, *p* = 0.006), and allele models (OR = 1.39, *p* = 0.016). Siddiqui et al. found an association of the TT allele of the *FGFR2* rs2981582 polymorphism with an increased BC risk in North Indian women [28]. Their study also reported the linkage of this variant with breast cancer risk in the premenopausal patient, where the T allele showed a stronger association with ER (+) and PR (+) clinicopathology history. The *FGFR2* rs2420946 polymorphism showed no significant association with breast cancer except in the overdominant model (aOR = 0.62, *p* = 0.048). A previous study conducted among Jewish and Arab Israeli BC populations also described the genetic variations of the *FGFR2* gene (including rs2981582 and rs2420946) as a risk factor for breast cancer [17].

Furthermore, several case–control studies were conducted in Mexican, West Siberian, South American, Chinese, Iranian, Australian Caucasian, and Jordanian Arab women to determine the correlation between *FGFR2* gene polymorphisms and breast cancer vulnerability [18,31,32,33,34,35,36]. They found that genetic variants in *FGFR2* (rs2981582 C > T, rs1219648 A > G, and rs2420946 C > T) were associated with breast cancer risk. Additionally, Liang Jie et al. demonstrated that breast cancer in Chinese women mainly occurred in the premenopausal stage [18]. However, another study with Chinese Han women showed the opposite result. Their findings showed that polymorphisms such as rs2420946, rs2981582, and rs1219648 were significantly linked with breast cancer risk in postmenopausal patients but not in premenopausal subjects [37]. They observed no correlation between hormone receptor status (estrogen and progesterone) and breast cancer risk.

Liang et al. conducted a case–control study in Southern Han Chinese women and confirmed the association of genetic variation of the *FGFR2* gene with the risk of breast cancer development [38]. Moreover, they stated that the association could differ with the variation of intrinsic subtypes. However, they found no significant correlations between *FGFR2* polymorphisms and ER/PR/HER2 subtypes of breast cancer [38]. Our study also found no significant association of ER/PR/HER2 hormone receptors with breast cancer susceptibility in the case of rs2981582 and rs2420946. Notably, a significant association of HER2 (+) receptor (*p* = 0.025) with BC risk was observed in the case of the rs1219648 variant. In addition, this variant also showed an increased risk of breast cancer with grade 2 cancer (*p* = 0.006) and positive lymph nodes (*p* = 0.043) in patients. Meanwhile, rs2420946 and rs2981582 only showed an association with N1 nodal status (*p* = 0.002) and positive lymph nodes (*p* = 0.004) accordingly.

Furthermore, the haplotype-based analysis demonstrated a significant association of GTT, ATT, and ATC haplotypes with breast cancer. In addition, this study illustrated the LD of all SNPs in the breast cancer cases and controls, with the LD between rs1219648 and rs2420946 (D’ = 0.931, r^2^ = 0.624); rs1219648 and rs2981582 (D’ = 0.777, r^2^ = 0.389), and rs2420946 and rs2981582 (D’ = 0.823, r^2^ = 0.606). The GEPIA and UALCAN databases revealed that breast cancer tissues possess greater levels of *FGFR2* mRNA expression than normal tissues (*p* < 0.01). Significantly lower expression of *FGFR2* in Asians (*p* = 2.83 × 10^−4^) was observed compared to Caucasians (*p* = 8.79 × 10^−1^). On the other hand, moderate expression was observed in African Americans (*p* = 2.09 × 10^−12^).

Notably, there were some limitations in our study that must be acknowledged and overcome in future research. Firstly, we only investigated three available SNPs of the *FGFR2* gene other than novel SNPs. Secondly, the comparatively small sample size of participants does not represent the whole scenario of all breast cancer patients in Bangladesh. Thirdly, there may be other factors, such as gene–environment interactions, which were not evaluated. Despite these limitations, this study has provided a clear indication of the relationship between *FGFR2* gene polymorphisms and the risk of breast cancer development in the Bangladeshi population.

## 5. Conclusions

Our study found that *FGFR2* genetic polymorphisms are significantly associated with breast cancer in Bangladeshi women. Different clinicopathological variables, such as HER2 status, cancer grade, and lymph node status, may have an impact on genetic polymorphisms that may be collectively associated with breast cancer risk. More research with a large sample size is suggested to confirm the findings of this study.

## Figures and Tables

**Figure 1 genes-14-00819-f001:**
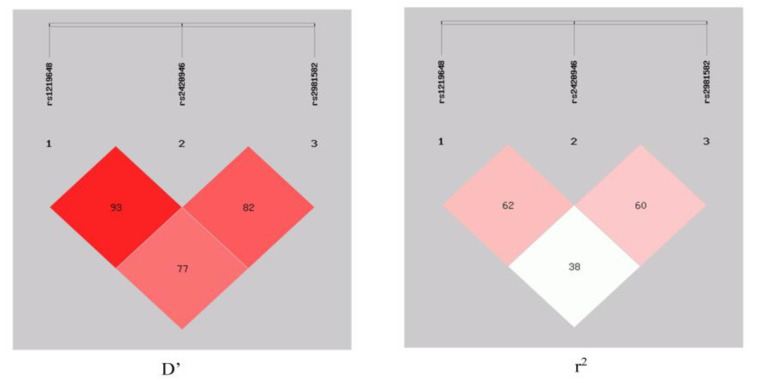
LD block between rs1219648, rs2420946, and rs2981582 in the case–control respondents.

**Figure 2 genes-14-00819-f002:**
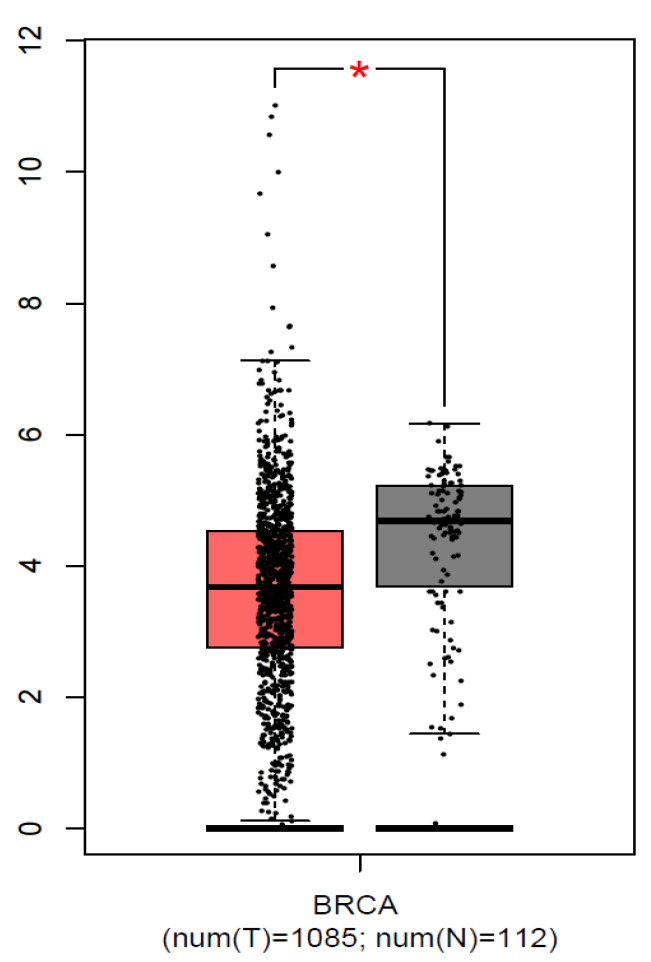
Expression of FGFR2 in breast cancer tissues and normal tissues. Each bar indicates the average level of expression of FGFR2. Error bars are the standard deviation of the mean value. Data from the GEPIA database were extracted (http://gepia.cancer-pku.cn/, accessed on 20 November, 2022). * statistical significance (*p* < 0.01).

**Figure 3 genes-14-00819-f003:**
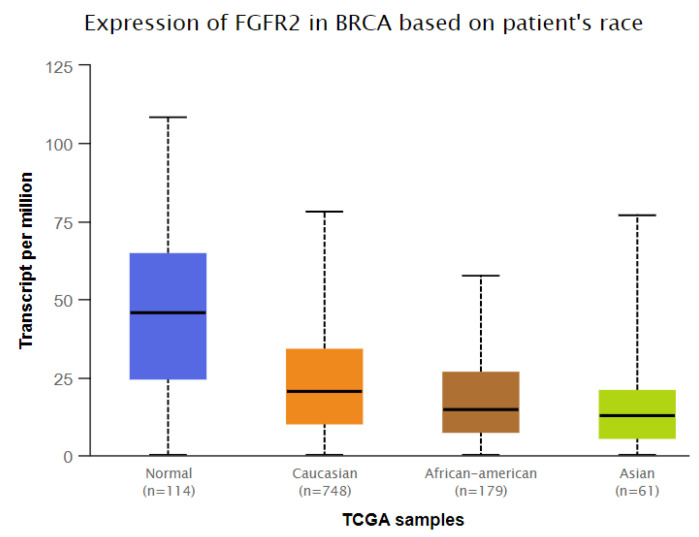
Association of *FGFR2* expression in different ethnic populations (Normal vs. Caucasian: *p* = 8.79 × 10^−1^, Normal vs. African American: *p* = 2.09 × 10^−12^, Normal vs. Asian: *p* = 2.83 × 10^−4^). Data from the UALCAN database were extracted (http://ualcan.path.uab.edu/, accessed on 20 November, 2022).

**Figure 4 genes-14-00819-f004:**
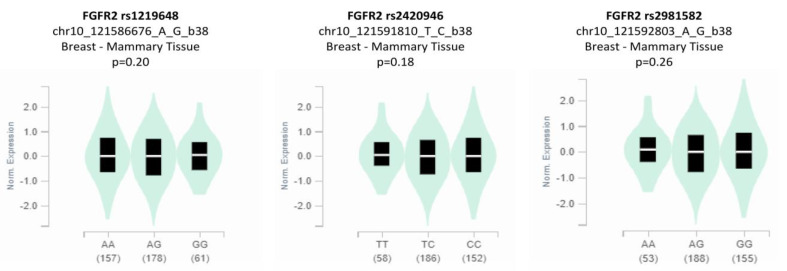
Genotype-based mRNA expression of *FGFR2* rs1219648, rs2420946, and rs2981582 polymorphisms.

**Table 1 genes-14-00819-t001:** Primers, PCR, and digestion conditions with respective fragment sizes.

SNP	Primer Sequences	PCR Conditions	No. of Cycles	Size of PCR Products	RE	Digestion Conditions	Digested Fragment Size (bp)
rs1219648A > G	FP: 5′-ACGCCTATTTTACTTGACACGC-3′ RP: 5′-GCTGGACAGGTCATTGTGGTG-3′	94 °C for 5 min 94 °C for 30 s 53 °C for 30 s 72 °C for 30 s 72 °C for 10 min 4 °C for ∞	35	230 bp	HinP1I	Incubation at 37 °C, 12 h.	NH: 230 (AA) HE: 230, 211, 19 (AG) MH: 211, 19 (GG)
rs2420946 C > T	FP: 5′-TTGGTGGAAGAGTCAGAAGA-3′ RP: 5′-GTGGAAAGGGACGAAGTT-3′	94 °C for 3 min 94 °C for 30 s 53 °C for 30 s 72 °C for 45 s 72 °C for 5 min 4 °C for ∞	35	429 bp	HinP1I	Incubation at 37 °C, 12 h.	NH: 107, 322 (CC) HE: 107, 322, 429 (CT) MH: 429 (TT)
rs2981582 C > T	F: 5′-CCCTTTGGAGACAACGTGAGCC-3′ R: 5′-CAGGCACCAGGTGGACTCTGC-3′	94 °C for 3 min 94 °C for 30 s 63 °C for 30 s 72 °C for 30 s 70 °C for 5 min 4 °C for ∞	35	176 bp	HinP1I	Incubation at 37 °C, 12 h.	NH: 22,154 (CC) HE: 22, 154, 176 (CT) MH: 176 (TT)

**Table 2 genes-14-00819-t002:** Demographic and clinicopathological features of breast cancer patients.

Parameters	Cases (226)	Controls (220)	*p*-Value
Age (Years)	42.89 (10.20)	41.41 (9.55)	0.114
≤35	69 (30.53)	85 (38.64)	0.072
>35	157 (69.47)	135 (61.36)	
Marital Status			
Married	220 (97.35)	208 (94.55)	0.142
Unmarried	6 (2.65)	12 (4.55)	
BMI			
Underweight (<18.5)	15 (6.64)	12 (5.45)	0.850
Normal (18.50–24.9)	77 (34.07)	83 (37.73)	
Overweight (25–29.9)	65 (28.76)	60(27.27)	
Obese (>29.9)	69 (30.51)	65 (29.55)	
Family History of Breast Cancer			
Yes	28 (12.39	15 (6.82)	0.049
No	198 (87.61)	205 (93.18)	
Smoking			
Never	223 (98.67)	218 (99.09)	0.677
Ever	3 (1.33)	2 (0.91)	
Age at Menarche			
≤13	161 (71.24)	158 (71.82)	
>13	65 (28.76)	62 (28.18)	0.892
Menopausal Status			
Continue	149 (65.93)	159 (72.27)	0.148
Stop	77 (34.07)	61 (27.73)	
History of OCP			
Yes	167 (73.89)	167(75.91)	0.624
No	59 (26.11)	53 (24.09)	
Alcoholism			
Yes	1 (0.44)	0 (0.00)	-
No	225 (99.56)	220 (100)	
Cases’ Clinicopathological Characteristics			
Estrogen Receptor Status		Grade of cancer	
ER (+)	88 (38.94)	Grade 1	62 (27.43)
ER (-)	138 (61.06)	Grade 2	131 (57.96)
Progesterone Receptor Status		Grade 3	33 (14.60)
PR (+)	86 (38.05)	Tumor size	
PR (-)	140 (61.95)	T0	4 (1.77)
HER2/neu Status		T1	53 (23.45)
HER2 (+)	60 (26.55)	T2	94 (41.59)
HER2 (-)	166 (73.45)	T3	34 (15.04)
Histological Type of Cancer		T4	41 (18.14)
Infiltrating Ductal Carcinoma	83 (36.73)	Lymph Node Status	
Invasive Ductal Carcinoma	129 (57.08)	Negative (-)	81 (35.84)
Others	14 (6.19)	Positive (+)	145 (64.16)
Nodal Status		Distant Metastasis	
N0	81 (25.84)	Mx	170 (75.22)
N1	100 (44.25)	M0	51 (22.57)
N2	36 (15.93)	M1	5 (2.21)
N3	9 (3.98)		

**Table 3 genes-14-00819-t003:** Genotype frequencies and association analysis of *FGFR2* rs1219648, rs2420946, and rs2981582 polymorphisms with breast cancer.

SNP ID	Model	Genotype/ Allele	Case (%)	Control (%)	HWE	Association Analysis	
						aOR (95% Cl)	*p*-Value
rs1219648		AA	84 (37.17)	132 (60.00)	0.054	1	
	Additive model 1 (AG vs. AA)	AG	100 (44.25)	70 (31.82)		2.87 (1.76–3.69)	<0.0001
	Additive model 2 (GG vs. AA)	GG	42 (18.58)	18 (8.18)		5.62 (2.52–12.54)	<0.0001
	Dominant model (AG + GG vs. AA)	AA	84 (37.17)	132 (60.00)		1	
		AG + GG	142 (62.83)	88 (40.00)		2.87 (1.76–4.69)	<0.0001
	Recessive model (GG vs. AA + AG)	AA + AG	184 (81.42)	202 (91.82)		1	
		GG	42 (18.58)	18(8.18)		4.04 (190–8.59)	<0.0001
	Overdominant model (AG vs. AA + GG)	AA + GG	126 (55.75)	150 (68.18)		1	
		AG	100 (44.25)	70 (31.82)		1.51 (0.93–2.46)	0.095
	Allele	A	268 (59.29)	334 (75.91)		1	
		G	184 (40.71)	106 (24.09)		2.16 (1.62–2.89)	<0.0001
rs2420946		CC	82 (36.28)	70 (31.82)	0.117	1	
	Additive model 1 (CT vs. CC)	CT	108 (47.79)	118 (53.64)		0.65 (0.38–1.11)	0.112
	Additive model 2 (TT vs. CC)	TT	36 (15.93)	32 (14.55)		1.30 (0.61–2.76)	0.499
	Dominant model (CT + TT vs. CC)	CC	82 (36.28)	70 (31.82)		1	
		CT + TT	144 (63.72)	150 (68.18)		0.77 (0.47–1.26)	0.302
	Recessive model (TT vs. CC + CT)	CC + CT	190 (84.07)	188 (85.45)		1	
		TT	36 (15.93)	32 (14.55)		1.57 (0.82–3.0)	0.173
	Overdominant model (CT vs. CC + TT)	CC + TT	118 (52.21)	102 (46.36)		1	
		CT	108 (47.79)	118 (53.64)		0.62 (0.39–1.0)	0.048
	Allele	C	272 (60.18)	258 (58.64)		1	
		T	180 (39.82)	182 (41.36)		0.94 (0.72–1.23)	0.640
rs2981582		CC	78 (34.51)	90 (40.91)		1	
	Additive model 1 (CT vs. CC)	CT	98 (43.36)	102 (46.36)	0.914	0.92 (0.54–1.57)	0.087
	Additive model 2 (TT vs. CC)	TT	50 (22.12)	28 (12.73)		2.60 (1.25–5.37)	0.010
	Dominant model (CT + TT vs. CC)	CC	78 (34.51)	90 (40.91)		1	
		CT + TT	148 (65.49)	130 (59.09)		1.23 (0.76–1.99)	0.408
	Recessive model (TT vs. CC + CT)	CC + CT	176 (77.88)	192 (87.27)		1	
		TT	50 (22.12)	28 (12.73)		2.47 (1.13–4.69)	0.006
	Overdominant model (CT vs. CC + TT)	CC + TT	128 (56.64)	172 (53.64)		1	
		CT	98 (43.36)	102 (46.36)		0.72 (0.45–1.16)	0.174
	Allele	C	254 (56.19)	282 (64.09)		1	
		T	198 (43.81)	158 (35.90)		1.39 (1.06–1.82)	**0.016**
	Haplotype	Cases	Controls	χ^2^		OR (95% CI)	*p*-Value
rs1219648, rs2420946, and rs2981582	ACC	0.481	0.510	0.720		0.91 (0.70–1.19)	0.502
	GTT	0.324	0.158	33.43		2.57 (1.86–3.55)	<0.0001
	ATT	0.029	0.144	37.69		0.18 (0.10–0.33)	<0.0001
	ACT	0.076	0.056	1.49		1.37 (0.80–2.34)	0.248
	GTC	0.038	0.062	2.54		0.61 (0.33–1.13)	0.115
	GCC	0.036	0.020	1.96		1.78 (0.78–4.08)	0.171
	ATC	0.006	0.049	15.44		0.13 (0.04–0.43)	0.0009

Bold values indicate statistically significant.

**Table 4 genes-14-00819-t004:** Associations of *FGFR2* rs1219648, rs2420946, and rs2981582 polymorphisms with clinicopathological variables.

Variables	Total	rs1219648		OR (95% CI)	*p*-Value	rs2420946		OR (95% CI)	*p*-Value	rs2981582		OR (95% CI)	*p*-Value
		AG + GG	AA			CT + TT	CC			CT + TT	CC		
Age													
≤35	68	42	26	1		41	27	1		48	20	1	
>35	158	100	58	1.07 (0.59–1.92)	0.828	103	55	1.23 (0.69–2.22)	0.483	100	58	0.72 (0.39–1.33)	0.291
BMI													
Underweight (<18.5)	15	9	6	0.91 (0.29–2.81)	0.865	10	5	0.85 (0.26–2.77)	0.791	7	8	0.45 (0.15–1.37)	0.157
Normal (18.50–24.9)	77	48	29	1		54	23	1		51	26	1	
Overweight (25.00–29.9)	65	32	33	0.59 (0.30–1.15)	0.118	33	32	0.44 (0.22–0.87)	0.019	37	28	0.67 (0.34–1.33)	0.256
Obese (>30)	69	53	16	2.00 (0.97–4.13)	0.061	47	22	0.91 (0.45–1.84)	0.793	53	16	1.69 (0.81–3.51)	0.161
Estrogen Receptor Status													
ER (−)	138	82	56	1		86	52	1		96	42	1	
ER (+)	88	60	28	1.46 (0.83–2.57)	0.185	58	30	1.17 (0.67–2.05)	0.584	52	36	0.63 (0.36–1.10)	0.107
Progesterone Receptor Status													
PR (−)	140	84	56	1		86	54	1		96	42	1	
PR (+)	86	58	28	1.38 (0.79–2.43)	0.262	58	28	1.30 (0.74–2.29)	0.362	52	36	0.63 (0.36–1.10)	0.107
HER2/neu Receptor Status													
HER2 (-)	166	97	69	1		110	56	1		111	55	1	
HER2 (+)	60	45	15	2.13 (1.10–4.13)	0.025	34	26	0.67 (0.36–1.22)	0.186	37	23	0.80 (0.43–1.47)	0.468
Histologic Type of Cancer													
Infiltrating Ductal Carcinoma	83	54	29	1		49	34	1		57	26	1	
Invasive Ductal Carcinoma	129	78	51	0.82 (0.46–1.46)	0.501	88	41	1.49 (0.84–2.64)	0.173	83	46	0.82 (0.46–1.48)	0.516
Others	14	10	4	1.34 (0.39–4.66)	0.643	7	7	0.69 (0.22–2.16)	0.528	8	6	0.61 (0.19–1.93)	0.399
Grade of Cancer													
Grade 1	62	48	14	1		43	19	1		43	19	1	
Grade 2	131	74	57	0.38 (0.19–0.75)	0.006	80	51	0.69 (0.36–1.32)	0.265	83	48	0.76 (0.40–1.46)	0.415
Grade 3	33	20	13	0.45 (0.18–1.12)	0.087	21	12	0.77 (0.32–1.89)	0.572	22	11	0.88 (0.36–2.18)	0.789
Tumor Size													
To	4	3	1	1		3	1	1		2	2	1	
T1	53	36	17	0.71 (0.07–7.30)	0.770	30	23	0.43 (0.04–4.46)	0.483	35	18	1.94 (0.25–14.97)	0.523
T2	94	60	36	0.56 (0.06–5.54)	0.617	61	33	0.62 (0.06–6.16)	0.680	58	36	1.61 (0.22–11.95)	0.641
T3	34	18	16	0.38 (0.04–3.98)	0.416	22	12	0.61 (0.08–6.54)	0.684	25	9	2.78 (0.34–22.75)	0.341
T4	41	27	14	0.64 (0.06–6.76)	0.713	28	13	0.72 (0.07–7.58)	0.783	28	13	2.15 (0.27–17.03)	0.467
Lymph Node Status													
Negative (-)	81	58	23	1		51	30	1		63	18	1	
Positive (+)	145	84	61	0.55 (0.30–0.98)	0.043	93	52	1.05 (0.60–1.85)	0.860	85	60	0.40 (0.22–0.75)	0.004
Nodal Status													
No	81	48	33	1		42	39	1		53	28	1	
N1	100	68	32	1.46 (0.79–2.69)	0.224	74	26	2.64 (1.42–4.93)	0.002	63	37	0.90 (0.49–1.66)	0.735
N2	36	21	15	0.96 (0.43–2.14)	0.925	23	13	1.64 (0.73–3.68)	0.228	26	10	1.37 (0.58–3.25)	0.470
N3	9	5	4	0.86 (0.21–3.44)	0.831	5	4	1.16 (0.29–4.64)	0.833	6	3	1.06 (0.25–4.55)	0.941
Distant Metastasis													
Mx	170	113	57	1		115	55	1		112	58	1	
M0	51	27	24	0.57 (0.30–1.07)	0.081	27	24	0.54 (0.28–1.02)	0.057	35	16	1.13 (0.58–2.22)	0.716
M1	5	2	3	0.34 (0.05–2.07)	0.240	2	3	0.32 (0.05–1.96)	0.218	1	4	0.13 (0.01–1.19)	0.070

## Data Availability

The data that support the findings of this study are available from the corresponding author upon reasonable request.

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
