# Peer review of "Evaluation of the Association between FGFR2 Gene Polymorphisms and Breast Cancer Risk in the Bangladeshi Population"

_genes, 2023, doi:10.3390/genes14040819_

Round 1

Reviewer 1 Report

In this study, the authors investigated the association of selected SNPs (rs1219648, rs2420946, and rs2981582) of the FGFR2 gene with breast cancer risk in Bangladeshi women. Although this topic is interesting, and the study is well designed, I have several concerns that should be addressed, as follows:

  • Methods:
    • The FGFR2 mRNA or IHC expression should be assayed and compared among different genotypes to determine the impact of FGFR2 SNPs on its expression level.
  • Results:
    • Many factors could affect the occurrence of breast cancer, such as age, gender, family history of breast cancer, smoking, alcoholism, age at menarche, menopausal status, history of OCP, etc., should be defined and compared between cases and controls in table 2 and be used to adjust the OR of each SNP in table 3.
    • HWE results should be specified for control in the table 3.
  • Conclusion: “Different clinicopatho-285 logical variables like HER2 status, grade of cancer, and lymph node status may be associated with breast cancer risk.” This statement should be revised to include the effect on polymorphisms.
  • Minor English revision is required.

Author Response

Comments and Suggestions for Authors

In this study, the authors investigated the association of selected SNPs (rs1219648, rs2420946, and rs2981582) of the FGFR2 gene with breast cancer risk in Bangladeshi women. Although this topic is interesting, and the study is well designed, I have several concerns that should be addressed, as follows:

Methods:

The FGFR2 mRNA or IHC expression should be assayed and compared among different genotypes to determine the impact of FGFR2 SNPs on its expression level.

Response:

Thank you very much for your valuable comment. We performed genotype-based FGFR2 mRNA expression for the studied SNPs from the Genotype-Tissue Expression (GTEx) database that has been added to the revised version of the manuscript.

Results:

Many factors could affect the occurrence of breast cancer, such as age, gender, family history of breast cancer, smoking, alcoholism, age at menarche, menopausal status, history of OCP, etc., should be defined and compared between cases and controls in table 2 and be used to adjust the OR of each SNP in table 3.

Response:

Thank you very much for your comment. The information of age, gender, family history of breast cancer, smoking, alcoholism, age at menarche, menopausal status, and history of OCP of both cases and controls have been added in the manuscript and compared these parameters between the cases and the controls (Table 2). The odds ratio has been adjusted for these parameters and provided in Table 3.

HWE results should be specified for control in table 3.

Response:

HWE p-values have been added in table 3

Conclusion: “Different clinicopatho-285 logical variables like HER2 status, grade of cancer, and lymph node status may be associated with breast cancer risk.” This statement should be revised to include the effect on polymorphisms.

Response:

This section has been modified according to the reviewer’s suggestion.

Minor English revision is required.

Response:

The manuscript has been checked thoroughly, and grammatical errors have been corrected.

Reviewer 2 Report

The manuscript needs some major revisions, as follow:

- The innovation of this article needs to be improved.

- It is best to avoid using abbreviations and acronyms in the abstract.

- The entire manuscript would benefit significantly from a grammatical revision.

- The authors should have considered the genetic ancestry as the potential confounders in the statistical analysis.

-  Start your discussion with introducing your aim and then discuss the clinical significance of your experiment. Then discuss findings, followed by your study limitations and then conclusion.

- In discussing your findings, compare and contrast them with other studies in the literature and develop arguments and hypotheses for your findings.

- In addition to previous research aligning with yours, please critically discuss those in disagreement and develop arguments and hypothesize. You may also add recommendations.

Author Response

Comments and Suggestions for Authors

The manuscript needs some major revisions, as follow:

- The innovation of this article needs to be improved.

Response:

Thank you very much for this valuable comment. This manuscript has been modified to improve its quality.

- It is best to avoid using abbreviations and acronyms in the abstract.

Response:

The modification has been performed.

- The entire manuscript would benefit significantly from a grammatical revision.

Response:

The entire manuscript has been checked with an English grammar expert and also checked with the professional version of  Grammarly software and modified to reduce grammatical errors.

- The authors should have considered the genetic ancestry as the potential confounders in the statistical analysis.

Response:

Modification has been performed by adjusting odds ratios with different potential confounders like demographic factors, clinicopathological and environmental factors to confirm that the association was solely due to the genetic variation.

-  Start your discussion with introducing your aim and then discuss the clinical significance of your experiment. Then discuss findings, followed by your study limitations and then conclusion.

Response:

Thank you very much for your crucial comment. Modifications have been made accordingly.

- In discussing your findings, compare and contrast them with other studies in the literature and develop arguments and hypotheses for your findings.

Response:

Thank you very much for your valuable comment. Modifications have been made accordingly.

- In addition to previous research aligning with yours, please critically discuss those in disagreement and develop arguments and hypothesize. You may also add recommendations.

Response:

Thank you very much for your valuable comment. Modifications have been made accordingly.

Round 2

Reviewer 1 Report

The authors have adequately addressed all my concerns.

Author Response

Response To the Reviewers:

Reviewer 1: The authors have adequately addressed all my concerns.

Response: Thank you very much for your comments.

Reviewer 2 Report

It is necessary that the authors described the detail of bands length for SNPs with added a PCR product figure.

A part of the samples must be sequenced and shown in figure.

Author Response

Response to the Reviewers:

2nd Reviewer

It is necessary that the authors described the detail of bands length for SNPs with added a PCR product figure.

Response: Thank you very much for your comments. The restriction enzyme digested gel images of the three studied SNPs (rs2981582, rs2420946 and rs1219648) have been added as supplementary materials (Figures S1-S3).
